# Strengthening Anomaly Awareness

Adam Banda[a], Charanjit K. Khosa[b], and Verónica Sanz[c]

[a]Southern Methodist University, Lyle School of Engineering, Dallas, 75205,TX, USA
[b]Department of Physics and Astronomy, University of Manchester, Manchester M13 9PL, United Kingdom
[c]Instituto de Física Corpuscular (IFIC), Universidad de Valencia-CSIC, E-46980 Valencia, Spain

### Abstract

We present a refined version of the Anomaly Awareness framework for enhancing unsupervised anomaly detection. Our approach introduces minimal supervision into Variational Autoencoders (VAEs) through a two-stage training strategy: the model is first trained in an unsupervised manner on background data, and then fine-tuned using a small sample of labeled anomalies to encourage larger reconstruction errors for anomalous samples.

We validate the method across diverse domains, including the MNIST dataset with synthetic anomalies, network intrusion data from the CICIDS benchmark, collider physics data from the LHCO2020 dataset, and simulated events from the Standard Model Effective Field Theory (SMEFT). The latter provides a realistic example of subtle kinematic deviations in Higgs boson production. In all cases, the model demonstrates improved sensitivity to unseen anomalies, achieving better separation between normal and anomalous samples. These results indicate that even limited anomaly information, when incorporated through targeted fine-tuning, can substantially improve the generalization and performance of unsupervised models for anomaly detection.

## 1 Introduction

The search for new physics beyond the Standard Model (BSM) remains a central objective of particle physics. A wide array of phenomena—from high-momentum tails at colliders to subtle deviations in particle couplings—could hint at underlying new dynamics. However, the vast landscape of BSM possibilities motivates search strategies that go beyond fixed signal hypotheses and are capable of identifying anomalies in a model-independent fashion.

Machine learning (ML) has emerged as a powerful tool to enhance discovery potential across particle physics [1,2]. Supervised ML techniques are widely used to discriminate between known signal and background events in collider analyses and direct detection experiments [3]. In the context of dark matter (DM) searches, ML methods have been applied

to both collider-based mono-$X$ topologies [4] and low-threshold direct detection experiments [5–8]. Nevertheless, these approaches depend on detailed prior knowledge of the signal, limiting their applicability to unforeseen scenarios.

This motivates growing interest in *anomaly detection* techniques, which aim to identify deviations from expected Standard Model processes without requiring specific signal labels. Unsupervised approaches, such as autoencoders [9–11] and variational autoencoders (VAEs) [12], train solely on background data and flag events with large reconstruction errors as anomalous. Normalizing flows provide an alternative based on density estimation, learning the probability distribution of Standard Model data and flagging low-likelihood events as potential signals [13, 14].

Weakly-supervised strategies offer a compromise between full supervision and complete model agnosticism. Techniques like Classification Without Labels (CWoLa) [15, 16], iterative tagging methods [17], and mass-agnostic classifiers [18–20] demonstrate improved sensitivity to subtle signals while avoiding strong signal-model assumptions. Hybrid strategies, such as Simulation Assisted Likelihood-Free Anomaly Detection (SALAD) [21], guide unsupervised learning using simulated background distributions, while adversarial [22] and semi-supervised [23] frameworks explore robust representations.

Flexible anomaly detection is also relevant in precision coupling measurements and effective field theory (EFT) interpretations, where subtle kinematic effects may arise from higher-dimensional operators [24–26]. Recent work has applied ML-based likelihoods [24] and unbinned multivariate techniques [25] to improve sensitivity in global SMEFT fits.

In parallel, new unsupervised anomaly detection benchmarks [27, 28] such as ANODE and Outlier Exposure explore realistic anomaly generalization scenarios in ML applications beyond HEP.

Despite these advances, key challenges remain. Unsupervised models may confuse rare but known Standard Model fluctuations with new physics, while weak supervision often assumes signal-like structures that may not match reality. Moreover, interpreting anomaly scores statistically remains an open issue [29].

In this work, we propose an *Anomaly Awareness* (AA) learning framework that combines the breadth of unsupervised learning with the focus of limited supervision. Building on the concept introduced in Ref. [30], we train models that are explicitly aware of anomalous events through a two-stage procedure: first, a VAE is trained unsupervised on background data; second, the model is fine-tuned using a small number of labeled anomalies, with a loss term that penalizes their accurate reconstruction.

We validate this strategy across multiple domains: digit classification (MNIST with Fashion-MNIST and CIFAR-10 anomalies), cyberattack detection using the CICIDS 2017 dataset, collider anomaly detection in the LHCO2020 dataset, and an EFT-inspired case study involving Standard Model Effective Field Theory (SMEFT) modifications to Higgs production. In this last case, we demonstrate how anomaly-aware VAEs can identify subtle deviations in collider kinematics caused by new physics operators. We further explore the impact of the anomaly penalty strength ($\lambda_{AA}$) on model performance, scanning a range of values and quantifying detection sensitivity through the area under the ROC curve (AUC).

The paper is organized as follows. In Section 2, we describe the refined Anomaly Awareness mechanism and its integration into Variational Autoencoders through a two-stage training strategy. Section 3 presents experimental validations across different domains, including image data, network traffic data, and high-energy physics datasets. Each experiment details the dataset preparation, model training, and evaluation metrics. Section 4 discusses the relevance of the results, tuning considerations, and potential future extensions.

## 2   Anomaly Awareness with Variational Autoencoders

Our approach builds upon the framework of Variational Autoencoders (VAEs) by introducing an Anomaly Awareness (AA) mechanism during training. A VAE consists of two neural networks: an encoder that maps an input $x$ into a latent variable $z$, and a decoder that reconstructs $x$ from $z$. The training objective of a VAE minimizes a loss function composed of two terms: a reconstruction loss, typically the mean squared error between the input and its reconstruction, and a Kullback-Leibler (KL) divergence term that regularizes the latent space to approximate a standard normal distribution.

Formally, the loss function for a sample $x$ is given by

$$\mathcal{L}_{\mathrm{VAE}}(x) = \mathbb{E}_{q(z|x)}\left[\|x - \hat{x}\|^2\right] + D_{\mathrm{KL}}(q(z|x)\|p(z)),$$

where $q(z|x)$ is the encoder distribution, $p(z)$ is the prior, and $\hat{x}$ is the reconstructed input. The reconstruction term ensures the fidelity of the reconstructed data, while the KL divergence imposes a smooth, continuous latent space.

The VAE architecture we employ consists of an input layer matching the dimensionality of the preprocessed features, followed by a fully connected hidden layer with 16 units and a ReLU activation. The latent space is two-dimensional, representing the mean and variance vectors needed for the reparameterization trick. The decoder mirrors the encoder structure with a hidden layer of 16 units (ReLU activation) and an output layer restoring the original feature dimension. The network is optimized using the Adam optimizer with a learning rate of $10^{-3}$ and a batch size of 512.

### 2.1   Anomaly Awareness Fine-tuning

Following unsupervised pretraining, we introduce Anomaly Awareness by fine-tuning the VAE with a modified objective. During fine-tuning, batches contain primarily background samples but are augmented with a small fraction (approximately 10%) of labeled anomalies. To encourage the model to produce higher reconstruction errors for anomalies while maintaining low errors for background samples, we modify the loss function.

The combined loss function becomes

$$\mathcal{L}(x) = \begin{cases} \mathcal{L}_{\mathrm{VAE}}(x), & \text{if } x \text{ is a background sample,} \\ -\lambda_{\mathrm{AA}}\mathcal{E}(x), & \text{if } x \text{ is an anomaly,} \end{cases}$$

---

**Algorithm 1** Two-Stage Training with Anomaly Awareness (AA)

---

**Require:** Background dataset $\mathcal{D}_{\mathrm{bg}}$, Anomaly dataset $\mathcal{D}_{\mathrm{anom}}$
**Ensure:** Trained VAE model with enhanced anomaly detection
 1: **Initialize** VAE parameters $\theta$
 2: **Set** maximum anomaly penalty $\lambda_{\mathrm{max}}$
 3: {Stage 1: Unsupervised Pretraining}
 4: **for** number of epochs **do**
 5:     Sample batch $B \subset \mathcal{D}_{\mathrm{bg}}$
 6:     Compute VAE loss $\mathcal{L}_{\mathrm{VAE}}(B)$
 7:     Update $\theta$ by minimizing $\mathcal{L}_{\mathrm{VAE}}(B)$
 8: **end for**
 9: Save pretrained model parameters
10: {Stage 2: Fine-tuning with Anomaly Awareness}
11: **for** epoch $e = 1$ to $E$ **do**
12:     Set $\lambda_{\mathrm{AA}} = \lambda_{\mathrm{max}} \times \frac{e}{E}$
13:     **for** each batch **do**
14:         Sample background batch $B_{\mathrm{bg}} \subset \mathcal{D}_{\mathrm{bg}}$
15:         Sample small anomaly batch $B_{\mathrm{anom}} \subset \mathcal{D}_{\mathrm{anom}}$
16:         Form combined batch $B = B_{\mathrm{bg}} \cup B_{\mathrm{anom}}$
17:         Compute reconstruction errors for all samples in $B$
18:         Compute loss:

$$\mathcal{L}(B) = \mathbb{E}_{x \in B_{\mathrm{bg}}} \left[ \mathcal{L}_{\mathrm{VAE}}(x) \right] - \lambda_{\mathrm{AA}} \times \mathbb{E}_{x \in B_{\mathrm{anom}}} \left[ \mathcal{E}(x) \right]$$

19:         Update $\theta$ by minimizing $\mathcal{L}(B)$
20:     **end for**
21: **end for**
22: **return** Trained VAE model

---

where $\mathcal{E}(x)$ is the reconstruction error for sample $x$, and $\lambda$ is a penalty coefficient promoting poor reconstruction of anomalies. The negative sign ensures that minimizing the loss corresponds to maximizing the reconstruction error for anomalous samples.

The coefficient $\lambda$ is not fixed but increases linearly during fine-tuning, starting from zero and reaching a maximum value $\lambda_{\mathrm{max}}$ over a number of epochs. This gradual introduction of the anomaly penalty avoids abrupt changes in the model behavior and allows the VAE to adapt progressively.

Overall, the two-stage training procedure consists of an unsupervised VAE pretraining phase focused purely on modeling background data, followed by a fine-tuning phase where the model learns to explicitly separate anomalies in reconstruction space. The algorithm is described in Algorithm 1.

Note that in the original Anomaly Awareness approach for classification, anomalies were mapped to uniform predictions to induce maximum uncertainty. In the VAE context, since

the output is a reconstructed object rather than a class probability, the analogous goal is to increase the reconstruction error for anomalies. Thus, while the spirit of anomaly awareness is preserved, the technical implementation differs: instead of centering anomalies, we push their reconstructions to be as inaccurate as possible.

# 3 Experiments

To evaluate the effectiveness and generality of the Anomaly Awareness mechanism, we perform a series of experiments across different domains. Our goal is to assess whether a VAE, trained predominantly on normal data and fine-tuned with a small sample of labeled anomalies, can improve its ability to detect unseen anomalous events.

We consider four datasets that span different types of input features:

- Handwritten digits and images (MNIST, Fashion-MNIST, and CIFAR-10).

- Network traffic features for cyberattack detection (CICIDS 2017).

- High-energy physics collider hadronic event features (LHCO2020).

- Anomalous Higgs production at the LHC in association with a gauge boson (SMEFT).

For each dataset, the experimental methodology is consistent. We first train a VAE on background data without any anomalies. We then fine-tune the VAE using a small set of labeled anomalies, applying an Anomaly Awareness penalty that encourages the model to poorly reconstruct anomalous samples. Finally, we evaluate the model's performance on unseen anomaly samples not included during fine-tuning. Reconstruction error distributions, Receiver Operating Characteristic curves, and Area Under the Curve scores are used to quantify the separation between normal and anomalous samples.

The following subsections detail the experimental setup and results for each domain.

## 3.1 MNIST with Synthetic Anomalies

The MNIST dataset, consisting of $28 \times 28$ grayscale images of handwritten digits, is used as the source of normal data. To simulate anomalies during fine-tuning, we employ the Fashion-MNIST dataset, which contains grayscale images of clothing items with the same dimensions but semantically distinct content. To evaluate the generalization ability to unseen anomalies, we use images from the CIFAR-10 dataset, resized to $28 \times 28$ and converted to grayscale.

**Preprocessing.** All images are normalized to the $[0, 1]$ pixel range. Standardization is performed using the mean $\mu_{\text{train}}$ and standard deviation $\sigma_{\text{train}}$ computed from the MNIST training set:

$$x' = \frac{x - \mu_{\text{train}}}{\sigma_{\text{train}}}.$$

The CIFAR-10 images are resized and grayscaled during loading to match the MNIST image format.

**VAE Architecture.** The VAE model consists of an encoder and a decoder. The encoder flattens the input images and maps them through a fully connected layer of 400 units with ReLU activation. Two parallel linear layers then predict the mean vector $\mu$ and log-variance $\log \sigma^2$ for the latent variables, with a latent dimension of 20. Sampling in latent space is performed via the reparameterization trick:

$$z = \mu + \sigma \odot \epsilon, \quad \epsilon \sim \mathcal{N}(0, I).$$

The decoder reconstructs the input by mapping $z$ through a fully connected layer with 400 ReLU units, followed by an output layer with 784 units reshaped to $28 \times 28$. A final Sigmoid activation ensures output pixel values lie in $[0, 1]$.

**Training Procedure.** The VAE is trained in two phases as described in Section 2.

**Phase 1: Unsupervised Pretraining.** The model is trained on 10,000 MNIST images for 10 epochs, using a batch size of 128 and the Adam optimizer with a learning rate of $10^{-3}$. The loss minimized is the sum of the binary cross-entropy (BCE) reconstruction loss and the Kullback-Leibler (KL) divergence:

$$\mathcal{L}_{\text{VAE}} = \text{BCE}(x, \hat{x}) + D_{\text{KL}}(q(z|x)\|p(z)).$$

**Phase 2: Fine-tuning with Anomaly Awareness.** Fine-tuning is performed by mixing normal MNIST samples with anomalous Fashion-MNIST samples in each batch. Normal samples contribute to the standard VAE loss, while anomalies are penalized based on their reconstruction error. The fine-tuning loss becomes:

$$\mathcal{L} = \mathcal{L}_{\text{VAE}}(\text{normal}) - \lambda_{\text{AA}} \times \mathcal{E}(\text{anomalies}),$$

where $\mathcal{E}(x) = \sum_{i=1}^{784}(x_i - \hat{x}_i)^2$ is the total pixelwise reconstruction error, and $\lambda_{\text{AA}}$ is set to a maximum of 0.1. Fine-tuning is conducted for 5 epochs with the same optimizer and batch size as in pretraining.

**Evaluation.** After training, reconstruction errors are computed separately for normal and anomalous samples. For final evaluation, 2,000 randomly selected CIFAR-10 images (resized and grayscaled) serve as unseen anomalies, while 10,000 MNIST test images serve as normal samples. Reconstruction errors are computed using the total squared pixel error:

$$\mathcal{E}(x) = \sum_{i=1}^{784}(x_i - \hat{x}_i)^2.$$

Histograms of reconstruction errors are plotted to visualize the separation between normal and anomalous data. A ROC curve is constructed, and the AUC is computed to quantify anomaly detection performance.

The reconstruction error distributions for normal samples and anomalies, before and after fine-tuning with Anomaly Awareness, are shown in Figure 1. The effect of fine-tuning is to shift the reconstruction errors of anomalies to higher values, improving separation.

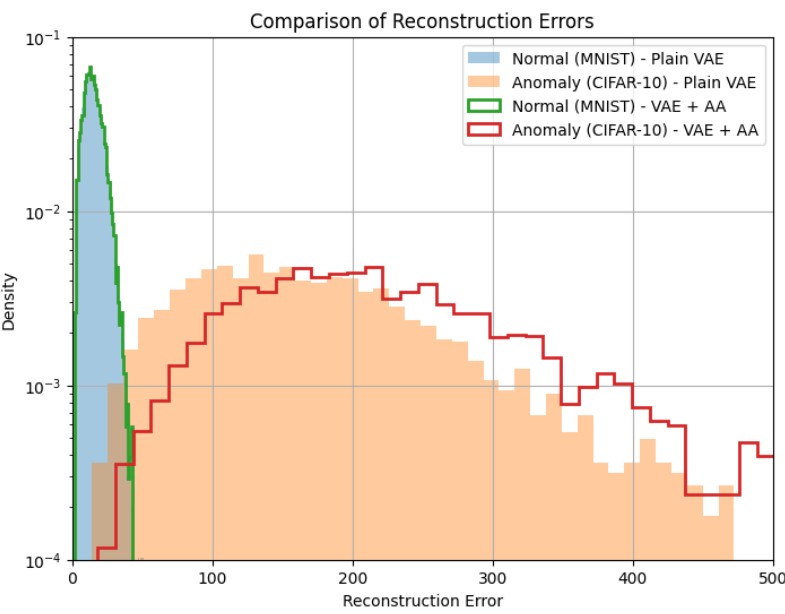

Figure 1: Reconstruction error distributions for normal (MNIST) and anomalous (CIFAR-10) samples, before and after fine-tuning with AA. Fine-tuning with AA shifts the anomalies towards higher reconstruction errors, improving separation.

The reconstruction error serves as the anomaly score: higher values indicate a higher likelihood of being anomalous. Reconstruction errors are computed for each sample as the sum of squared differences between the input and the reconstruction.

To evaluate performance, we compute the Receiver Operating Characteristic (ROC) curve by sweeping a threshold over the reconstruction error. For each threshold value, the model classifies a sample as anomalous if its reconstruction error exceeds the threshold. Normal samples (MNIST) and anomalous samples (CIFAR-10) are assigned binary labels of 0 and 1, respectively.

This allows computation of the True Positive Rate (TPR) and False Positive Rate (FPR) at each threshold, based on the known ground truth labels. The AUC summarizes this curve into a single scalar value, representing the model's ability to separate normal and anomalous samples across all possible thresholds. The AUC score is then obtained by integrating the ROC curve. An AUC of 0.5 corresponds to random performance, whereas an AUC closer to 1 indicates better separation between normal and anomalous samples. In this case the AUC for plain VAE is 0.9987 whereas for VAE plus AA tuning is 0.9997.

## 3.2 Cyberattack Detection with CICIDS 2017

To evaluate the performance of Anomaly Awareness on structured tabular data, we use the CICIDS 2017 dataset [31], a benchmark dataset designed for intrusion detection research. The dataset contains realistic network traffic captures, including benign traffic and various attack scenarios such as brute force attempts, denial-of-service (DoS) attacks, infiltration, and web attacks. Each flow is represented by a set of statistical features including packet lengths, inter-arrival times, flag counts, byte counts, and flow durations.

**Data Preprocessing.**   We use traffic captured on Tuesday and Wednesday of the dataset, corresponding mainly to benign activity, Brute Force SSH/FTP attacks, and DoS attacks. Unnecessary columns such as IP addresses, timestamps, and flow IDs are dropped. Labels are mapped into three classes: benign (background), Brute Force (anomaly for fine-tuning), and DoS (unseen anomaly for testing). Samples from other classes are discarded.

Features are preprocessed as follows: highly skewed features (skewness $> 1$ in the background class) are log-transformed using a $\log(1 + x)$ transform after clipping negative values to zero. Standardization to zero mean and unit variance is performed based only on the benign (background) samples.

**Training Procedure.**   The VAE architecture used here is lightweight, consisting of an input layer matching the feature dimension, a hidden layer of 16 ReLU units, a latent space of dimension 2, and a mirrored decoder. The VAE is pretrained on benign traffic samples for 50 epochs using a batch size of 512 and the Adam optimizer with a learning rate of $10^{-3}$.

Fine-tuning is performed by injecting a small proportion (10%) of Brute Force anomaly samples into each batch during training, using the Anomaly Awareness mechanism described in Section 2. The anomaly penalty coefficient $\lambda_{AA}$ is linearly increased up to 5.0 during the 20 fine-tuning epochs. During fine-tuning, per-sample anomaly reconstruction errors are clipped to avoid numerical instability.

**Evaluation.**   After fine-tuning, the VAE is evaluated on unseen DoS attacks. No DoS samples were seen during training or fine-tuning. Reconstruction errors are computed for normal and DoS samples, and ROC curves and AUC scores are used to quantify separation. The reconstruction error distributions for benign and DoS traffic, before and after fine-tuning with Anomaly Awareness, are shown in Figure 2.

Fine-tuning with Brute Force anomalies improves the model's ability to detect unseen DoS attacks, shifting anomalous samples towards higher reconstruction errors and increasing the AUC (0.79) compared to the baseline VAE (0.75).

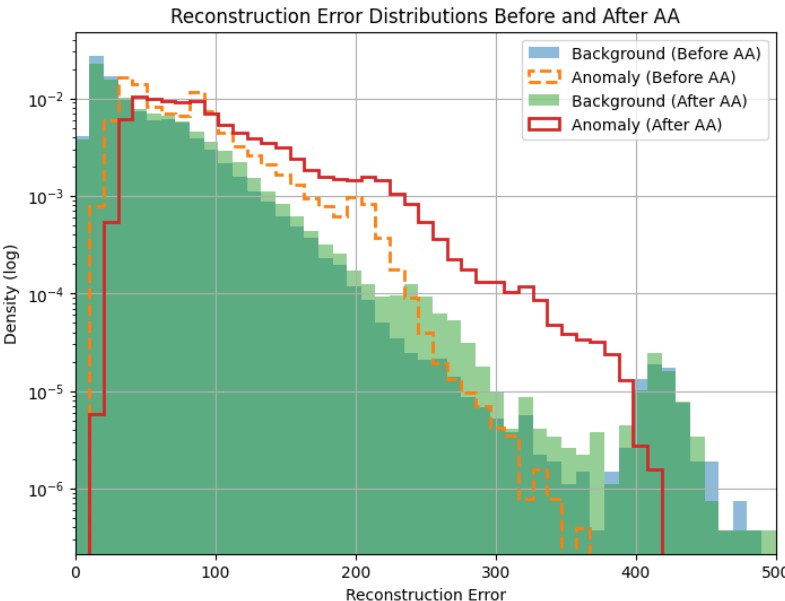

Figure 2: Reconstruction error distributions for benign traffic and DoS attacks, before and after fine-tuning with Anomaly Awareness. Fine-tuning with Brute Force anomalies shifts unseen DoS attacks to higher reconstruction errors.

## 3.3 Anomaly Detection in High-Energy Physics with the LHC Olympics 2020 anomaly detection challenge

To test the Anomaly Awareness method on collider physics data, we use the LHCO2020 dataset [32, 33], which contains simulated high-level features of jets from proton-proton collision events at the Large Hadron Collider (LHC). The dataset is designed for anomaly detection studies and includes both Standard Model background events and a set of signal events from a hidden sector model.

**Data Preparation.**  Each event is represented by 15 features, corresponding to kinematic and substructure observables of the two leading jets: the three-momentum components $(p_x, p_y, p_z)$, jet mass $m$, and $N$-subjettiness variables $\tau_1$, $\tau_2$, and $\tau_3$. The full dataset contains 1,100,000 events, with 1,000,000 labeled as background and 100,000 as signal.

The dataset is downloaded from Zenodo and loaded using pandas. Background and signal events are separated based on the `label` column. To correct for strong feature skewness, we apply a $\log(1 + x)$ transformation to features with skewness greater than 1, computed only on the background. Negative values are clipped to zero before the transform. The most skewed features include jet masses and $N$-subjettiness variables of both jets.

After the log transformation, all features are standardized to zero mean and unit variance using the statistics of the background. The same transformation is applied to the signal

data. The background events are split into 70% for training and 30% for testing, while the signal events are split into 70% for fine-tuning and 30% for final evaluation.

**Training.** The Variational Autoencoder architecture mirrors that used in the cyberattack experiment. It consists of a fully connected encoder and decoder with 16 hidden units and a two-dimensional latent space. The model is first pretrained on background samples for 50 epochs with a batch size of 128 and the Adam optimizer with a learning rate of $10^{-3}$. After pretraining, a fine-tuning phase with Anomaly Awareness is conducted. During fine-

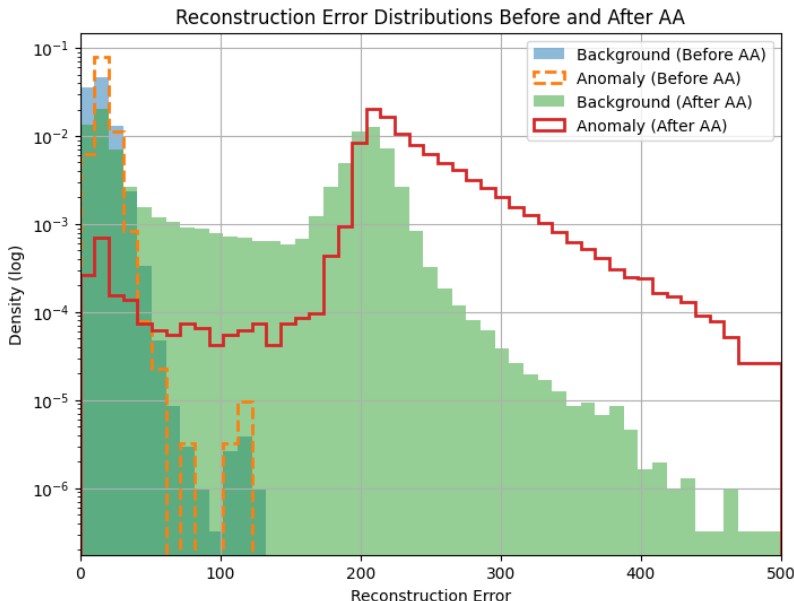

Figure 3: Reconstruction error distributions for background and signal events in the LHCO2020 dataset, before and after fine-tuning with Anomaly Awareness. Fine-tuning shifts signal events to higher reconstruction errors and improves separation.

tuning, approximately 10% of each batch is composed of signal samples injected alongside background samples. The anomaly penalty described in Section 2 is applied, with the penalty coefficient $\lambda$ linearly increased from zero to 1.0 over 10 fine-tuning epochs. To maintain numerical stability, per-sample reconstruction errors for anomalies are clipped at a maximum value of 200.

**Evaluation.** The model is evaluated on unseen background and signal events, which were not included during fine-tuning. Reconstruction errors are computed as the sum of squared differences between input and output features. Samples are labeled as 0 (background) and 1 (signal). Using the reconstruction error as the anomaly score, a ROC curve is constructed and AUC is computed.

Fine-tuning with Anomaly Awareness substantially improves anomaly detection performance: the AUC increases from 0.64 (standard VAE) to 0.90 (VAE with AA). The reconstruction error distributions for background and signal events, before and after fine-tuning, are shown in Figure 3. Fine-tuning shifts the signal events towards higher reconstruction errors, improving separation.

## 3.4 Anomaly Detection in SMEFT Higgs Production

The Standard Model Effective Field Theory (SMEFT) provides a systematic framework to parametrize potential effects of new physics through higher-dimensional operators built from Standard Model fields. This approach enables a consistent extension of the SM Lagrangian while respecting its symmetries, and has become a widely adopted tool for interpreting precision measurements at the LHC and other experiments.

This is currently a very active area of research, with global SMEFT analyses attempting to constrain dozens of operator coefficients simultaneously using hundreds of observables from Higgs, electroweak, and top quark sectors. State-of-the-art fits rely on complex pipelines that incorporate experimental data, theoretical predictions, and renormalization group evolution. Public tools such as `SMEFiT` [34] or `Fitmaker` [35] have been developed to facilitate and standardize this effort. These global analyses are powerful but computationally intensive and rely on precise modeling assumptions, motivating complementary approaches that can discover or characterize deviations in a more model-agnostic way.

Among these, dimension-six operators that modify the coupling of the Higgs boson to vector bosons are of particular interest. For example, we simulate the effect of one such operator

$$(D^\mu H)^\dagger \sigma^a (D^\nu H) W^a_{\mu\nu}, \tag{1}$$

which can significantly affect the kinematics of associated Higgs production processes.

In this study, we simulate the process $pp \to ZH$, where the Higgs boson decays to a pair of bottom quarks, and the $Z$ boson decays to a pair of charged leptons ($Z \to \ell^+\ell^-$, $H \to b\bar{b}$). This simulation is made with `MadGraph5_aMC@NLO` [36, 37]

We generate two datasets: one corresponding to Standard Model (SM) predictions, and one including Beyond the Standard Model (BSM) effects from the SMEFT operator above set to a large value, ensuring that this sample represents pure BSM contributions to the vertex.

Each dataset contains 200,000 events and 13 kinematic variables: the transverse momenta of the leading and subleading $b$-jets and leptons, $p_T$ of the reconstructed Higgs and $Z$, angular variables such as pseudorapidity ($\eta$) and azimuthal angle ($\phi$) of the reconstructed bosons, the angular distance between the two leptons, and the invariant mass of the full system (denoted $\hat{s}$).

We apply the same anomaly detection pipeline as in previous experiments: a VAE is trained on SM data alone, and then fine-tuned using a small number of BSM examples via Anomaly Awareness.

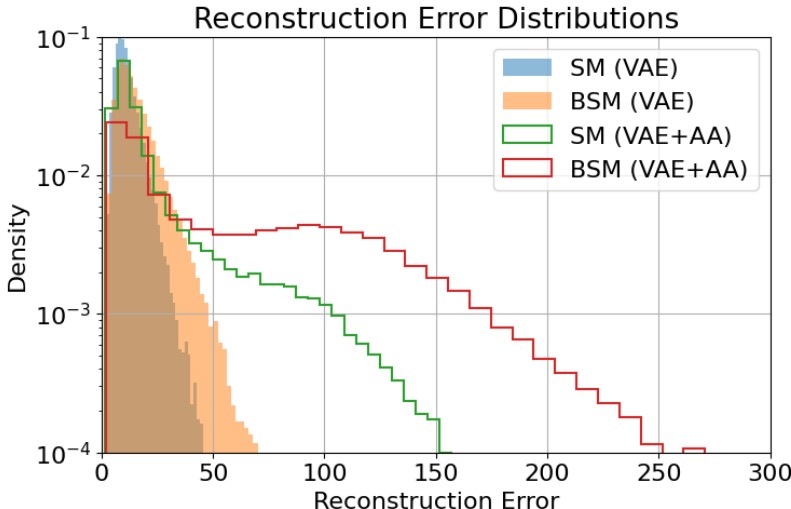

Figure 4: Reconstruction error distributions for SM and BSM events in the SMEFT case, with and without Anomaly Awareness fine-tuning.

Figure 4 shows the distribution of reconstruction errors for SM and BSM events, before and after the AA fine-tuning. The AA procedure shifts the BSM distribution to higher values, improving separability. The corresponding ROC curve confirms this behavior: the AUC improves from 0.64 to 0.71 when AA is applied for a fixed maximum value of $\lambda_{AA} = 0.07$.

Note that in general care must be taken when setting the anomaly penalty coefficient $\lambda_{AA}$ in the loss function. Large values of $\lambda_{AA}$ led to degradation of the reconstruction quality for SM events, reducing the model's ability to distinguish SM from BSM. We found that a small value, $\lambda_{AA} \simeq 0.1$, provided a good balance: it increased the reconstruction error for BSM events while leaving the SM reconstruction largely unaffected. This behaviour is illustrated in Fig. 5.

# 4 Discussion and Outlook

In this work, we presented a refined version of the Anomaly Awareness (AA) mechanism, originally proposed to enhance anomaly detection in unsupervised models [30]. Unlike the original AA approach, which introduced anomalies as a distinct distributional prior, our method incorporates anomalies directly during training in a two-stage procedure: pretraining on background-only data followed by fine-tuning with anomaly-aware supervision.

The improvement lies in the way anomalies are introduced: rather than altering the model's latent space assumptions, we encourage the model during fine-tuning to assign systematically larger reconstruction errors to known anomalies. This approach remains minimally supervised, requiring only a small subset of labeled anomalies, and does not

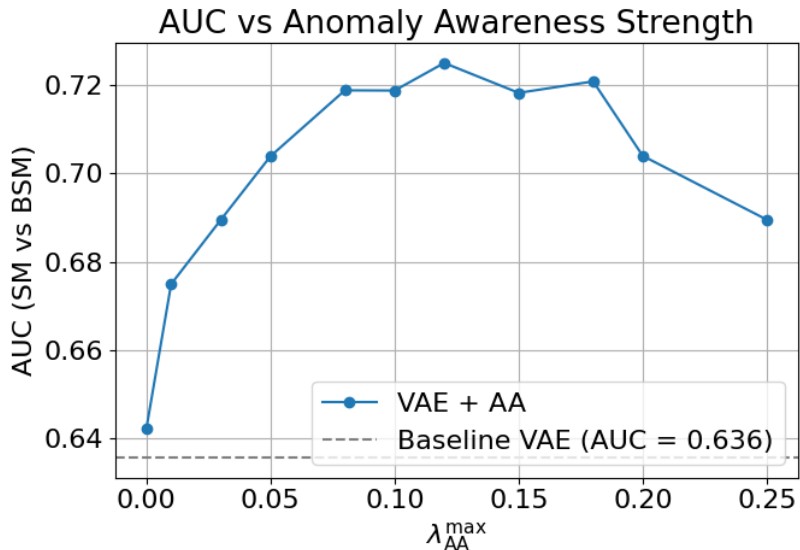

Figure 5: Variation of AUC as a function of the value of $\lambda_{\text{AA}}$.

modify the unsupervised objective during the initial training phase.

Our experiments demonstrate that this method improves anomaly detection performance across very different domains. We tested the method on image data (handwritten digits versus clothing items), structured tabular data from cyberattacks (benign traffic versus network intrusions), and high-energy physics collider data (background jets versus signal jets). In each case, the Anomaly Awareness fine-tuning led to improved separation between normal and anomalous samples, as reflected by increases in AUC scores.

Importantly, we evaluated the models on unseen anomalies, different from those used during fine-tuning. This provides evidence that the method generalizes and does not merely overfit to the particular anomalies seen during training. The consistent improvement across datasets with different input structures and semantic meanings suggests that the approach is robust.

While our results are encouraging, future work could further explore the sensitivity of the method to the amount and type of anomaly data used during fine-tuning, as well as its application to more complex datasets and models.

Overall, the results indicate that the refined Anomaly Awareness mechanism is a simple yet effective way to strengthen the performance of unsupervised anomaly detection methods, even when only limited anomaly information is available.

## Acknowledgements

The research of VS is supported by the Proyecto Consolidacion CNS2022-135688 from the AEI and the Ministerio de Ciencia e Innovacion project PID2023-148162NB-C21, and the

*Severo Ochoa* project CEX2023-001292-S funded by MCIU/AEI. Work of CKK is supported by the Royal Society through the Newton Fellowship Alumni follow on funding.

This project made use of computing resources provided by the IFIC cluster and the University of Manchester computing facilities.

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
