# Peer review of "Strengthening Anomaly Awareness"

_SciPost Physics_

## Round 1 · Referee Report · Anonymous (Referee 1) · 2025-6-25

Strengths

1- The paper proposes a simple modification to Variational Auto-Encoders to enhance the sensitivity to anomalies; 2- The experiments include datasets from several domains; 3- All the experiments show enhanced out-of-distribution detection of unseen anomalies.

Weaknesses

1- It is unclear how, in each experiment, a proper analysis can make use of anomaly awareness training; 2- The description of the neural network architecture and the training methodology lack some clarity; 3- The proposed modification to the training loss is simple but, in the current description, is not theoretically well-founded; 4- The effect of the exposure to the anomalies used during training is rather limited for the HEP experiments; the impact on the background distribution, as well as other synthetic anomalies, is not well described (see Report and Requested changes)

Report

The authors propose a less-unsupervised approach for autoencoder-based anomaly detection. A standard VAE is first trained on background events. Then, a second training step explicitly encourages larger reconstruction errors for a small set of anomalies injected in the training set.

While other methods in the literature already propose a similar modification founded on the minimization of a likelihood loss, this work does not include a similar theoretical foundation in its current form.

Although I appreciate the inclusion of experiments from different domains, the inclusion of the MNIST and CICIDS datasets seems unusual for this journal and outside its covered specialties.
Given the need for text clarifications together with extra results, the paper should undergo a major revision before I can consider it ready for publication.

Requested changes

1- The introduction to VAEs could be improved. For instance, the reader should be able to find the definition of the encoder distribution p(z|x) and the analytic form of the KL term in the text. I am assuming it is a Gaussian, but this will increase the clarity of the paper; 2a- Please consider clarifying the neural network architecture used in the paper; Sec.2 presents a first description, however, different architectures are then reported in each subsection in Sec.3; 2b- The previous comment also applies to the loss function; there are two L_VAE loss functions; one uses an MSE reconstruction while the second one computes the BCE; 3a- The selection of the value of \lambda_{AA} should be studied for all the experiments. The authors provide a scan for the SMEFT application, but it is unclear if it will also apply to the other datasets; 3b- A similar concern applies to the fraction of anomalous events used in each batch;

More general comments: 4- Why did the authors use MNIST instead of image datasets closer to physics? For instance, jet images are a well-explored data format for anomaly detection. The authors could consider using such a dataset. This would also allow for a comparison with other anomaly detection methods in the literature. 5- How do the authors define "unseen anomalies" for the LHCO and SMEFT experiments? It is clear that for MNIST and CICIDS the unseen anomalies are from a different out-of-distribution set, while for the remaining two experiments these unseen anomalies are not well defined. 6- Fig.2 and Fig.3 show that the background distribution is significantly distorted by the anomaly awareness term. Currently, no explanation is provided on how the distinction between background and potential signal would be performed on real unlabeled data.

Recommendation

Ask for major revision

---

## Round 1 · Referee Report · Anonymous (Referee 2) · 2025-7-4

Strengths

  1. The authors apply their method to four different datasets from different domains, showing that their method is applicable to more than a single area.
  2. The paper is very clearly written and easy to follow.

Weaknesses

  1. The authors only quote the AUC as a performance metric. It would have been good to see some plots like ROC curves or SIC curves.
  2. I miss a discussion and comparison of the different results on the different datasets, since both the original AUC and the performance increase varies.

Report

The authors present an interesting study, where sensitivity to anomalies are enhanced in a VAE model through the use of small labeled datasets in a finetuning step. They show the performance on four different datasets, and in all four cases the sensitivity to anomalies is enhanced through their method. As far as I am aware, this is a novel approach in the physics anomaly detection literature and fits well with the chosen Journal.

Requested changes

  1. In the first two cases (MNIST and Cyber attacks), the model is finetuned on anomalies of one type and then evaluated on anomalies of a different type. I would like to see some comment, or even feature plots, on how different these anomalies are to one another, and also to the background.

  2. In the last two cases (LHCO and SMEFT), the model is finetuned on the same type of anomalies as it is then evaluated on, albeit different data points. In the case of LHCO, it would be interesting to see how well the model does when finetuned on the R&D dataset (which is what I assume is being done in the study, please clarify this in the text), and then applied to one (or more) of the black box datasets which are also included in the LHCO challenge. This would correspond to a more realistic scenario, and also be in the spirit of the original challenge.

  3. In the LHCO scenario, if you do use the black boxes it would be very interesting to see how well the model does with the different types of anomalies. I do think such a study would benefit the paper, and likely open up for a discussion of the scope and limitations of this method.

  4. It would be interesting to see the ROC curves mentioned in the text, for all four cases.

  5. The choice of lambda (penalty parameter) and the number of epochs required in the finetuning step differ between the four datasets. No explanation is given for how these parameters are chosen. It would be interesting to comment on this, especially on whether the lambda parameter needs to be set very carefully (as is mentioned in the SMEFT case) and what that means for the generalizability of the method.

  6. In figure 3, showing the reconstruction error for the LHCO dataset, the reconstruction errors for both the background and the anomalies increase dramatically. In comparison, in the MNIST case and the cyber attack case, the change is not this dramatic. In particular, the reconstruction error for the background seems to not change at all in the MNIST case. Do you understand why we see this behavior? Could it have something to do with the original separation power? In the MNIST case, the anomalies were already very anomalous with an AUC of 0.9987.

  7. Please specify the amount of signal injected in the fintetuning step in the MNIST case and the SMEFT case. For the other two scenarios, you have used a 10% mixture. What happens if you change this number?

  8. "In parallel, new unsupervised anomaly detection benchmarks [27, 28] such as ANODE and Outlier Exposure explore realistic anomaly generalization scenarios in ML applications beyond HEP" - I'm missing the refs for ANODE and Outlier Exposure here

  9. Check the capitalization of the titles in the reference list.

  10. Please provide refs to the datasets used in the MNIST case.

  11. Please provide a ref to where the SMEFT dataset can be downloaded, or provide more information if needed to enable its reproduction.

Recommendation

Ask for minor revision

---

## Editorial Decision

awaiting_resubmission